# Design and implementation of a robust data logging and satellite telemetry system for remote research

Sunil N. Oulkar\*,<sup>1</sup>, Matthew W. Peacey<sup>2</sup>, Michael Mitrev<sup>3</sup>, Duncan J. Quincey<sup>1</sup>, Bryn Hubbard<sup>2</sup>, Tom Matthews<sup>4</sup>, Ankita S. Oulkar<sup>2</sup>, Katie E. Miles<sup>5</sup>, Ann V. Rowan<sup>6</sup>

- <sup>1</sup>School of Geography, University of Leeds, Leeds, United Kingdom
  <sup>2</sup>Centre for Glaciology, Department of Geography and Earth Sciences, Aberystwyth University, Aberystwyth, United Kingdom
  <sup>3</sup>Ground Control Technologies UK Ltd.
  - <sup>4</sup>Department of Geography, King's College London, London, United Kingdom
- 5Lancaster Environment Centre, Faculty of Science and Technology, Lancaster University, Lancaster, United Kingdom
  - <sup>6</sup>Department of Earth Science, University of Bergen and Bjerknes Centre for Climate Research, Bergen, Norway
- 15 \*Correspondence to: Sunil N. Oulkar (S.N.Oulkar@leeds.ac.uk)

#### Abstract




Scientific research frequently requires data acquisition and transmission from remote environments, requiring robust, autonomous solutions capable of operating in extreme environmental conditions with minimal maintenance. This study presents the design and implementation of a data logging and telemetry system deployed in the Western Cwm of Mount Everest, Nepal, to transmit several meteorological parameters from an automatic weather station and firn layer temperatures obtained from a suite of borehole thermistors. Drawing on recent advances in satellite Internet of Things (IoT) connectivity, we present the successful integration and deployment of Campbell Scientific data loggers with Ground Control's compact satellite-enabled RockREMOTE Mini, which uses the Iridium Certus 100 networks and is powered by Iridium's 9770 modem. The complete system, which operated at 6,660 m a.s.l, in an extremely cold climate with a limited sky-view factor due to the steep surrounding terrain, provided continual monitoring of ice temperatures and meteorological conditions transmitted every 24 hours, from May 4 to August 10, 2025. Data integrity and transmission reliability were consistently maintained despite the harsh weather conditions and limited power availability. This integrated system established a robust methodological framework for other researchers working in remote locations, demonstrating the potential for sustained and high temporal resolution measurements of environmental conditions in locations where traditional communication infrastructure is unavailable.






35 **Keywords:** Data Logging, Telemetry System, CR1000, RockREMOTE Mini, Automatic Weather Station (AWS), Thermistor Strings, Remote Research, Real-Time Monitoring

#### 1. Introduction

The cryosphere is crucial for many Earth surface and climatic processes, regulating Earth's energy balance and global temperatures, and influencing weather patterns, ocean circulation, sea level and seasonal water storage (Fountain et al., 2012; Hieronymus & Kalén, 2020; Kulkarni et al., 2021). However, the cryosphere is changing at an accelerating rate (Hugonnet et al., 2021), with many glaciers predicted to disappear by the end of the century (Rückamp et al., 2025). Understanding the processes that govern mass and energy exchange across glacier surfaces and within subsurface firn environments is essential for predicting the future response of polar and high-mountain regions to climate change (Gardner et al., 2023; Hay & Fitzharris, 1988), but capturing these interactions reliably over time and across space remains a significant challenge. This is particularly the case in remote environments such as the Arctic, Antarctic, and high-mountain regions, where logistical access is severely limited and communication infrastructure varies greatly and is, in some cases, effectively non-existent, with only satellite communication available.

Current approaches to remote environmental monitoring rely primarily on periodic manual data collection and the deployment of equipment including Automatic Weather Stations (AWS). Various glaciological monitoring programmes have documented the difficulty of capturing *in situ* measurements of glacier dynamics, often requiring multiple field visits per season or year to maintain or modify equipment, retrieve data, and address sensor failure (e.g., Costanza et al., 2016; Martin et al., 2014; Martinez & Hart, 2010; Matthews et al., 2020; Miles et al., 2018; Oulkar et al., 2024; Pernov et al., 2024; Tartari et al., 2009; Tetzlaff et al., 2017). Such expeditions incur substantial costs, involve safety risks, and present logistical complexities, which could be avoided with the availability of a data logging system equipped with satellite telemetry capabilities.

We implemented an integrated data logging and satellite telemetry system specifically designed for deployment in extreme high-altitude environments. The system combines Campbell Scientific CR1000/CR1000X data loggers with Ground Control's RockREMOTE Mini satellite communication units, based on Iridium Certus 100 connectivity to provide reliable data transmission from locations where terrestrial communication infrastructure is absent. We





demonstrate the framework on which our system is built so that others can replicate, adapt, and improve this design for their autonomous environmental monitoring purposes. We first present the design of the monitoring system, then describe its performance with an emphasis on the power and data management strategies used. We conclude with proposing telemetry protocols that are suitable for remote and extreme research settings.

#### 2. Study site and data specification

The integrated data logging and telemetry system was deployed in the Western Cwm of Mount Everest, Nepal, at an elevation of 6,660 m a.s.l. (27.97°N, 86.93°E), as part of a broader project to study near-surface energy exchanges and ice temperatures in the accumulation zone of one of the world's highest glaciers (Fig. 1). The Western Cwm contains the upper section of the Khumbu Glacier, which flows west from the Lhotse Face through the Khumbu Icefall, before turning south and terminating 5-10 km farther down-valley (RGI 7.0 2023). The Western Cwm contains a series of mountaineering camps (Camp I, Camp II and Camp III) used for climbing Mt. Everest via the southeast ridge in Nepal and providing access to the high peaks of Lhotse and Nuptse (Figs. 1a and 1b). Despite having infrastructure in place to service the mountaineering community, deploying cryospheric monitoring equipment in the Western Cwm is both dangerous and expensive; in particular, fieldwork is challenging because of the extreme temperatures, the narrow weather window in which the site can be accessed, and the numerous mountain hazards presented by the navigating the icefall with its associated avalanche and rockfall risks, risk of heat and cold injury, and working at high altitude where low oxygen levels increase the likelihood of acute mountain sickness (Matthews et al., 2020). Equipment deployment is further complicated by limited helicopter support, the need for climbing permits, physical limits on carrying loads, and the need for Sherpa expertise. The development of an autonomous data logging and transmission system that reduced the need for regular field visits therefore had multiple benefits to the project team and reduced the environmental footprint of the research project.

The monitoring system included an AWS and two subsurface thermistor strings, connected to Campbell Scientific CR1000 and CR1000X data loggers respectively, each with an integrated RockREMOTE Mini satellite communication unit (Fig. 1c). Boreholes for the installation of thermistors were drilled into the firn layer to a total depth of 12 m using a Heucke Steam Drill (Heucke, 1999). The AWS incorporated meteorological sensors including air temperature,


relative humidity, snow depth, incoming shortwave radiation and incoming and outgoing longwave radiation.

The system was powered by 12 V 24 Ah sealed lead acid batteries with 30 W lightweight solar panel charging arrays mounted vertically on the support mast to maximise solar energy collection throughout the day. Data loggers and telemetry units for the AWS were housed in robust weatherproof cases and in Campbell Scientific enclosures for the thermistor arrays, mounted on glass-reinforced plastic poles installed 1.5 m into the surface. The RockREMOTE Mini satellite communication system was integrated with each data logger through a serial communication interface.

Figure 1. (a) Khumbu Glacier and the surrounding area, shown by satellite image (ArcPro basemaps (Maxar)) (b) The field site, located in the Western Cwm of Mount Everest, Nepal, at an elevation of 6,660 m a.s.l. (27.97° N, 86.93° E), and (c) fully deployed integrated monitoring system showing the complete telemetry enabled setup including AWS, thermistor borehole, solar power, CR1000/CR1000X enclosures, RockREMOTE Mini and satellite communication antenna for Iridium Certus 100 connectivity. Note that the cables were trimmed and secured to the mounting poles before final deployment.





#### 3. Data logging and telemetry system

#### 3.1. Data logger: Campbell scientific CR1000 and CR1000X series

The Campbell Scientific CR1000 and CR1000X series data loggers have become the standard for environmental monitoring in extreme conditions due to their reliability, extensive sensor compatibility, and flexible programming capabilities. They have been successfully deployed in the polar regions, high-altitude environments, and in a range of other challenging locations (Citterio et al., 2015; Doyle et al., 2018; Li et al., 2018; Sicart et al., 2014; Sugiyama et al., 2015), including at the Bishop Rock close to the summit of Mt. Everest (Matthews et al., 2022). The ability of the CR1000 and CR1000X loggers to operate across a wide temperature range (-40°C to 70°C) and their robust construction make them particularly suitable for deployment in harsh conditions, while their comprehensive analogue and digital input capabilities allow for integration with a wide variety of sensors commonly used in remote research. Additionally, the system's compatibility with multiple communication protocols, including satellite communication, offers the possibility of data transmission. The CR1000/CR1000X loggers operate using the CRBasic programming language, which can accommodate complex data collection protocols, including conditional sampling based on environmental conditions, data quality control algorithms, and power management strategies. This functionality enables customised data collection and processing according to the specific research requirements of the user.

#### 3.2. Telemetry system: RockREMOTE Mini

The RockREMOTE Mini, powered by Iridium's 9770 modem, is a compact, efficient, and robust version of the RockREMOTE family that operates on the Iridium Certus 100 network, developed by Ground Control Technologies United Kingdom Ltd (Ground Control: RockREMOTE Mini, 2025). Its physical components comprise a RockREMOTE Mini device, a SCAN antenna (65020-011), an Registered Jack-45 (RJ45) connector, Category 5 Enhanced (CAT-5E) ethernet cable, a combined power and serial communication General-Purpose Input/Output (GPIO) cable, 9 m of coaxial (LMR 400) antenna cable with N plug and TNC plug, and a bespoke antenna mounting kit (Fig. 2). Its compact form (182 × 74.5 × 53 mm) and robust IP66 rated enclosure are designed to make the device resilient to the harshest of weather conditions, including extended exposure to temperatures ranging from -40°C to +70°C and relative humidity ≤ 95%. The RockREMOTE's ultra-low operating power consumption of 300

(C) (I)



mW, combined with a sleep pin for dynamic power management, is ideally suited to solar-powered or battery-constrained deployments in remote environments.

The RockREMOTE Mini supports Iridium Certus 100 with Internet Protocol (IP) connectivity (up to 88 Kbps downlink, 22 Kbps uplink) and IMT (Iridium Message Transport), enabling both standards-based Power over Ethernet (PoE) integration via IP and scalable, efficient data transport via IMT. The omnidirectional connectivity eliminates the need for precise antenna pointing, a significant advantage in the dynamic harsh environment where equipment positioning may shift due to glacier movement. Its versatile interface options, including Ethernet with PoE+, RS232/RS485 serial communication, and GPIO connectivity, enable integration with a wide range of sensor networks and data acquisition systems. The device's built-in Global Navigation Satellite System (GNSS) receiver provides a reliable time source for connecting equipment, while, Ground Control's cloud platform (Cloudloop), enables data integration and storage (see section 7).

Figure 2. Telemetry system components: (a) RockREMOTE Mini device, (b) RockREMOTE SCAN antenna (65020-011), (c) antenna mounting kit, (9d) m LMR-400 cable with N plug and TNC plug connectors and (e) GPIO cable. Source: <a href="https://www.groundcontrol.com/product/rockremote-mini/">https://www.groundcontrol.com/product/rockremote-mini/</a> (last access: 27 November 2025). Reproduced with permission.





#### 3.3. Integration of data logger with RockREMOTE Mini

The CR1000/CR1000X and RockREMOTE Mini have compatible communication interfaces and protocols (Fig. 3). The CR1000/CR1000X native support for serial communication (RS232) aligns with the RockREMOTE Mini interface connectors, enabling their integration without requiring additional hardware adapters or protocol converters. The CR1000/CR1000X CRBasic programming environment allows users to integrate AT command syntax by embedding the required ASCII prefix format (AT+CMD) followed by the payload and CRLF, facilitating direct communication with the RockREMOTE Mini and utilising the IMT protocol for efficient, low bandwidth data transmission. Power management is achieved through the CR1000/CR1000X digital control outputs, which can directly interface with the RockREMOTE Mini sleep pin for intelligent power cycling based on environmental conditions or data collection schedules. The CR1000/CR1000X built-in timing capabilities complement the RockREMOTE Mini GNSS receiver, ensuring precise data timestamping and synchronisation across multiple sensor networks. This compatibility extends to data formatting, where the CR1000/CR1000X flexible data processing capabilities can optimise payload structures for IMT message-based transmission, maximising the efficiency of satellite airtime.

Figure 3. Architecture of the remote monitoring system showing the integration of data logging, telemetry, power supply, and sensor components for autonomous operation in remote environments. Note: The middle four connection lines between the CR1000 and RockREMOTE Mini are GPIO cable. Adapted from <a href="https://www.groundcontrol.com/blog/integrating-rockremote-mini-with-cr1000/">https://www.groundcontrol.com/blog/integrating-rockremote-mini-with-cr1000/</a> (last access: 27 November 2025). Reproduced with permission.




#### 3.4. Data logger programming

CRBasic provides the functionality to be able to handle multi-sensor data acquisition, implement intelligent power management, and design efficient satellite communication protocols optimised for remote monitoring (Fig. 4). The system implements the RockREMOTE Mini's IMT protocol support through serial communication at 115,200 baud, using the message-based approach that eliminates protocol overhead by transmitting data in Base64 format with no header information, thereby requiring substantially lower bandwidth compared to IP-based communication.

Data acquisition and storage intervals can be programmed flexibly; for the Western Cwm deployment we scanned all raw data at the high-frequency interval of 10 s and saved average, minimum, and maximum values at lower frequency of 1,800 s (30 minutes). While all data were written to the data logger memory for long-term storage, only the low frequency 1,800 s values were transmitted via the telemetry system. This dual resolution approach ensured the measurement and recording of rapid environmental variability while maintaining manageable data volumes for satellite transmission. The CRBasic code to achieve this is available at <a href="https://doi.org/10.5281/zenodo.16985625">https://doi.org/10.5281/zenodo.16985625</a>.

The RockREMOTE Mini sleep pin provides dynamic power control, with the program automatically activating the satellite modem three minutes before scheduled transmissions to ensure communication stability while minimising power consumption during dormant periods. This interval is user-configurable and can be adjusted based on specific deployment requirements or environmental conditions. The transmission protocol implements a 24-hour data collection cycle with primary transmission scheduled at 11:02 local time, followed by systematic retry attempts every hour (i.e., at 12:02, 13:02, and 14:02) if initial transmission fails (Fig. 4). This schedule aligns with peak solar radiation periods to ensure the batteries are optimally charged during transmission attempts. In cases where daily data transmission fails completely, the system implements a cumulative data buffering approach where nontransmitted data are appended to the following day's dataset. The accumulated data follow the same scheduled transmission protocol the next day at 11:02, with systematic retry attempts if required. The system can continue accumulating and transmitting multi-day datasets for 4 days, with the transmission capacity ultimately limited by the size limits of the satellite communication protocol (100 KB) or the character limit; which was set to a string limit of 25,000 characters based on the available memory size in the logger. As new data are appended




at the start of the string, characters beyond the limit are truncated, ensuring that the most recent data are always retained. The program also incorporates subroutines that manage message tracking, status verification, and automatic retry mechanisms, with each transmission attempt generating a unique message identifier that denotes its success or failure status.

The string parsing and data formatting protocols of RockREMOTE Mini implement the AT command syntax required for IMT (messaging size up to 100 KB per message) communication (Ground Control: Mini Serial Interface, 2025), with the program dynamically constructing transmission strings that include timestamp information, sensor data with appropriate precision formatting, and delimiting characters that enable efficient server-side data parsing. AT+IMTWU (MO IMT in Binary Mode without CRC, 100 KB per message) commands initiate binary data transmission with dynamic length specification based on the formatted data string. Message status tracking utilises the IMT protocol's built-in acknowledgement system, with the program monitoring response codes to determine transmission success and implementing automatic message cancellation and retry protocols when transmission failures are detected. AT+IMTMOS (IMT MO Status) commands are used to query transmission state through response code parsing, with successful transmissions indicated by state code "5", while failed attempts trigger AT+IMTC3 message cancellation commands followed by automatic retry sequences, as described above.

Figure 4. Workflow of the CRBasic program for the CR1000X, showing sensor initialisation, data acquisition, averaging, storage, and Iridium transmission with retry and power management logic. Panel Temp is panel temperature, Temp is air temperature, RH is relative humidity, SWin is incoming shortwave radiation, LWin is incoming longwave radiation, and LWout is outgoing longwave radiation and resistance is thermistor resistance.

# 4. Power management



Power management represents a critical design challenge in high-altitude deployments where solar charging is limited by weather conditions and seasonal variation in illumination. Our system implements an energy management strategy that seeks to balance system performance with power conservation requirements. The CRBasic program transmits only when battery voltage is  $\geq 11.8$  V, and the transmission process shuts down completely at voltages that are  $\leq 11.5$  V to prevent deep discharge damage to the battery. Figure 5 shows battery voltages for




all three of our deployed loggers during the period of operation, ranging with strong diurnal cyclicity (reflecting solar charging) between 12.5 and 14.5 V.

Figure 5. The 30-minute minimum recorded battery voltage during the monitoring period for the three lead-acid batteries powering the deployed loggers (one for the AWS and two for the thermistor strings), indicating power system performance under remote deployment conditions.

# 5. Interface: Cloudloop IoT platform

The Cloudloop Internet of Things (IoT) platform serves as the interface through which data transmitted by the RockREMOTE Mini can be accessed (Fig. 6a). The data received from the RockREMOTE Mini arrive in Base64 format, but each message can be visualised as HEX and ASCII within the Cloudloop interface (Fig. 6b). The platform distinguishes between different message types (e.g. start-up routines, successful data receipt, failed transmissions, routine logs) and provides detailed payload information for each successful transmission, as well as logging precise timestamps, message sizes, and transmission success rates, so that data integrity and system performance can be tracked. Cloudloop's onward delivery methods are Email, Azure, AWS, Google Storage, FTP, HTTP Webhook and MQTT, +ThingSpeak and ThingsBoard.

In our implementation, Email serves as the primary channel for data transfer for extraction. The system is configured to dispatch an email on receipt of every new message, which has proved to be a simple but reliable mechanism for near real-time data forwarding without requiring complex API integrations. This provides an effective means for conducting daily quality checks and facilitating the integration of our data into downstream processing pipelines.

Figure 6. Cloudloop IoT dashboard showing (a) telemetry data from the RockREMOTE Mini with timestamps, type, message size, and payload (b) Detailed message in the HEX and ASCII format (Basemap from Leaflet, © OpenStreetMap).

# 275 6. Automated ETL pipeline for telemetry data processing

To automate the extraction, transformation, and loading (ETL) of telemetry data received via Cloudloop's Email delivery method, we implemented a Microsoft Power Automate-based workflow that eliminates manual intervention in the satellite-to-analysis pipeline (Fig. 7).




Alternative approaches include custom scripts in Python or R, cloud-based services such as AWS Lambda, or other automation platforms like Zapier, depending on user preference and expertise. Once Cloudloop receives data from the RockREMOTE Mini and sends an automated email containing the HEX payload, the Power Automate flow is triggered (Fig. 8). The flow monitors the inbox for incoming messages from Cloudloop and retrieves the required HEX payload from the email body. The system then reads the existing HEX data file to retrieve previously stored observations, appends the new payload, and saves the combined dataset as a text file in a designated location, preserving continuity across transmission cycles. Following this ETL sequence, an Excel workbook automatically reads the updated text file, converts the HEX encoded content into ASCII format, and loads the decoded data into a structured dataset for analysis. The data size limits to be considered are only those of Excel workbook limits. The whole process is completed within tens of seconds, meaning the data are accessible in analysisready format within ~1 minute from the workflow being initiated at the monitoring site. Small delays can arise during data transformation and refresh time within the Excel workbook, which can be tackled by directly using the HEX file with Python or similar programming languages. The Power Automate flow and associated files available are at https://doi.org/10.5281/zenodo.16985625.

Figure 7. Remote monitoring system architecture schematic illustrating the complete data pathway, starting from RockREMOTE Mini, then transmitting via Iridium satellite, followed by processing on the Cloudloop platform, subsequent email delivery, integration with Microsoft Power Automate ETL workflow, and concluding with the final processed dataset. Adapted from <a href="https://www.groundcontrol.com/blog/integrating-rockremote-mini-with-cr1000/">https://www.groundcontrol.com/blog/integrating-rockremote-mini-with-cr1000/</a> (last access: 27 November 2025). Reproduced with permission.

Figure 8. Power Automate workflow for automated extraction, transformation, and loading of HEX telemetry data into an Excel-ready format.

#### 7. Discussion



## 7.1. System performance and reliability

The integrated data logging and telemetry system, combining the CR1000/CR1000X data logger with the RockREMOTE Mini Iridium (Encompassing the Iridium 9770 transceiver), proved to be robust and reliable throughout its operation, successfully transmitting data from its deployment on May 4, 2025, until August 10, 2025, when measurements ceased due to structural collapse of the AWS and thermistor setup. While this failure has not yet been investigated in the field, it was likely caused by surface melting, which may have displaced the solar panel and disrupted power, rather than by any malfunction of the telemetry system itself. The system's resilience during extensive cloud cover, typical of the monsoon season, has been

particularly noteworthy, with successful data transmission maintained even during periods of heavy overcast conditions that typically challenge satellite communication systems. Indeed, transmission data show that across all three setups (AWS and two thermistor strings), 273 out of 293 total transmission attempts were successful on the first attempt, representing a 92% first-attempt success rate. Of the remaining 20 transmissions, 17 were transmitted on the second attempt, and the remaining 3 were transmitted on the subsequent day (Fig. 9). Notably, the next-day transmission protocol achieved a 100% success rate for any data that failed to transmit the previous day on all four daily attempts, ensuring complete data recovery without any data loss throughout the entire deployment period.


Figure 9. Data transmission performance of the AWS and both thermistor datasets, showing daily data delivery attempts. Blue bars indicate successful first attempts, purple bars indicate second attempts, red bars indicate no transmission and grey bars shows successful transmissions on the following day first attempts.


Similarly, the battery system performed reliably throughout the operating period (Fig. 5), with no sign of voltage drop or power failure. However, it is important to note that during the winter season reduced solar radiation and lower temperatures would present a more significant challenge to maintaining the battery charge.


The CR1000/CR1000X data logger was selected for its robustness, sensor compatibility, and cold-environment reliability, but other data acquisition systems, including low-cost platforms like Raspberry Pi (Nazir et al., 2017), Arduino (Chan et al., 2021), Cryologger Ice Tracking Beacon (ITB) (Garbo & Mueller, 2024), HOBO RX3000, Vaisala AWS310 and DL-Series or







commercial microcontrollers, may also be sufficient for simpler deployments or where rapid prototyping is needed. These systems may be preferable under different conditions, including application-specific constraints such as environmental severity, required precision, sampling frequency, maintenance feasibility, cost, robustness, ability to operate across large temperature ranges, power demand, and acceptable deployment duration.

Satellite modem alternatives such as MetOcean's STREAM series also provide comparable telemetry capabilities and may be selected depending on application-specific criteria such as power constraints, data volume, latency tolerance, operational costs, support availability and integration needs. Further, the current deployment relied on the RockREMOTE Mini modem operating over the Iridium satellite network due to its global coverage and low power demand, but several alternative satellite communication options are also available and may be more suitable for different use cases. Platforms such as Thuraya and Inmarsat offer regional geostationary coverage with higher bandwidth but typically may require more power and lineof-sight stability, making them more effective at lower-latitude and less obstructed environments. Other satellite communication platforms include ARGOS, which offers global coverage but with limited positional precision, and Globalstar, which provides cost-effective communication but lacks reliable coverage at high latitudes (Garbo & Mueller, 2024). Starlink offer high-bandwidth data transfer and expanding global coverage through a growing constellation of low-Earth-orbit satellites. However, such platforms may currently require significantly higher power and infrastructure. Additionally, access may be restricted in certain regions due to geopolitical or regulatory limitations. As a result, the use of these platforms in autonomous remote monitoring remains constrained both technically and operationally.

While the upfront cost of the telemetry system, including hardware and Iridium communication, ranges from £1,700–2,000 (\$2,300–2,600) per site, and data plans cost £20-30 (\$27–37) per month, these expenses are outweighed by efficiency savings on travel to remote environments. Manual data retrieval campaigns in remote regions often exceed £1,000–5,000 (\$1,350–6,750) per person for alpine or Arctic expeditions, with greater costs for Antarctic and high-mountain expeditions, varying depending on the specific region and conditions. These expeditions often require extensive logistics, helicopter access, permits, specialist equipment, and in-country personnel, incurring further costs. Additional complications can arise from extreme weather, technical terrain, exposure to illness, and



physical and mental strain, in some cases leading to aborted trips where the data cannot be retrieved.

The nature of the environments that the sensors are deployed in can also lead to data loss from destruction of equipment. This can occur from avalanches, equipment melt-out, loggers falling into crevasses and destruction from local wildlife (Immerzeel et al., 2014). In some deployments, sites may not be safely visited at all after installation (e.g., on fast-moving ice, in an icefall, on an iceberg), making satellite telemetry the only practicable method of data retrieval. Such telemetry-based systems not only eliminate the need for repeat access but also reduce the risk of complete data loss in the event of equipment failure or loss. Additionally, the ability to monitor equipment status in real-time enables early fault detection, potentially allowing a rapid response to resolve the issue, and maximises data recovery thus improving the scientific return. Indeed, although our installation is no longer functioning, the data are stored in the cloud, within the automated email receipts, and locally as part of the Power Automate workflow, rather than solely on the loggers that may or may not be retrievable during the next fieldwork season.

## 7.2. Future perspective and implications

The current system's limitation to IMT protocol, while highly effective for the present application, presents an opportunity for future enhancement. Implementing dual protocol capability, combining IMT for routine data transmission with IP connectivity for higher bandwidth applications, would expand the system's capabilities significantly. This hybrid approach would enable remote system diagnostics, firmware updates, and transmission of higher resolution data products when conditions permit, while maintaining the cost effective IMT protocol for standard operations.

Future deployments could consider implementing more sophisticated sensors for monitoring system health, including communication signal strength logging. These additional data streams would provide valuable insights into system performance under varying environmental conditions and enable predictive maintenance strategies.

The success of this system underscores its value as a model for standardised deployment in other data-sparse cryospheric regions. For example, extending this architecture to monitor surge-type glaciers in the Karakoram, the Andes, or the Alaska Range could support real-time monitoring of glacier dynamics where field visits are challenging. In regions like the Tibetan

Plateau, European Alps and Svalbard, similar deployments could provide critical insights into active layer thawing, permafrost variability, and surface energy exchanges with a changing climate. Furthermore, beyond cryospheric applications, this telemetry framework has broad potential across multiple domains of remote monitoring. With minimal changes, the system could support a range of alternative payloads including time-lapse cameras, data to describe hydrological parameters, seismic signals (Maurer et al., 2020) and geotechnical (i.e. borehole) measurements. Furthermore, this technology has the potential to contribute meaningfully to early warning systems for geotechnical and climatic hazards in remote regions. The modularity of the system makes it adaptable for regional or national monitoring frameworks, enabling scalable networks that can provide timely data to both researchers and decision-makers across cryospheric, marine, terrestrial, and atmospheric monitoring domains.

## 8. Summary remarks





The successful implementation and operation of an integrated data logging and telemetry system in the Western Cwm of Mount Everest demonstrates the feasibility of continuous, high-resolution environmental monitoring in extreme conditions. The system achieved exceptional reliability and data quality while demonstrating significant advances in communication efficiency and power management throughout its operating period.

The integration of Campbell Scientific CR1000/CR1000X data loggers with the Ground Control's RockREMOTE Mini, which has advanced satellite communication capabilities, provides a robust platform for autonomous environmental monitoring in locations where terrestrial communication infrastructure is unavailable. The communication approach successfully balanced operational efficiency with cost effectiveness, achieving significant reductions in communication costs while maintaining operational flexibility.

The power management strategies developed for this deployment demonstrate that continuous operation is achievable in extreme environments through careful system design and intelligent power management. The integration of dynamic power control for communication systems represents a particularly valuable innovation for resource-constrained applications.

Our programming scripts and automated data processing workflows are openly available to support reproducibility and broader adoption across the international research community, addressing the limitations of traditional manual monitoring systems and reducing the risk of data loss and operational downtime in the world's most challenging environments. Future work

should focus on expanding the sensor capabilities of the system and developing standardised deployment protocols that enable widespread adoption of this monitoring approach in remote research applications.

## Code availability

The CRBasic code and automated data extraction workflow are available at <a href="https://doi.org/10.5281/zenodo.16985625">https://doi.org/10.5281/zenodo.16985625</a>.

#### **Author contribution**




AVR, BH and DJQ conceptualised the study. SNO and TM developed the CRBasic code. MWP and BH implemented the code for thermistor integration. MWP, DJQ, and TM installed the equipment in Everest's Western Cwm, supported by BH, KM and SNO from Everest Base Camp. MM provided technical support for the RockREMOTE telemetry system. ASO developed the automation workflow for data extraction. All authors contributed to the discussion and preparation of the manuscript.

## Competing interests.

MM is employed by Ground Control, who design and distribute the RockREMOTE series of products.

## Acknowledgments

We thank Himalayan Research Expeditions and Seven Summits Trekking for invaluable help during the field season. Special thanks to the technical support provided by Ground Control for their assistance in system configuration. ChatGPT was used only to improve the readability of existing text.

## **Financial support**

This work was funded by a NERC Pushing the Frontiers grant (NE/Z000033/1) awarded to DJQ, BH and AVR, on which SNO and MWP are employed. KM was supported by a Royal Society Research Grant (RG\R1\251524), and KM and AVR were supported by the Mount Everest Foundation.

485

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
