# Peer review of "Design and implementation of a robust data logging and satellite telemetry system for remote research"

_EGUsphere, 2025_

## Referee Comment (RC1)

**Summary:** The paper presents an integrated data logging and satellite telemetry in extreme high-altitude conditions, combining Campbell data loggers with the RockREMOTE modem. While the introduction effectively motivates the need for environmental data, and the manuscript provides detailed descriptions of system components, programming, and automated data workflows, the novelty and technical justification for design/instrument choices are often unclear. Key limitations include the lack of quantitative evaluation of telemetry performance, power consumption, data transmission reliability, and error handling. Also, minimal critical assessment of alternative loggers or satellite platforms. Overall, the manuscript demonstrates a functional and thoughtfully engineered deployment with promising automation and power management strategies for remote environmental monitoring.

**Abstract**

1.  The title is generic and does not indicate what aspect of the system is novel.
2.  The motivation in the abstract does not explain what exact limitations in existing data logging or telemetry systems necessitated this work.
3.  It implies novelty using recent satellite IoT connectivity but does not explain why this integration represents an advance over previously available telemetry solutions. If this is the first time IoT has been used in this research domain, it would be helpful to highlight it.

**Introduction**

1.  Lines 51-60: it does not clearly state a specific technical problem it addresses (e.g. data transmission reliability, power limits, data loss, or system lifetime), instead describing a general need for remote monitoring.
2.  The literature focuses on field logistics and maintenance, with little discussion of existing satellite telemetry systems, making the novelty of the work unclear. It is not evident whether the system is an incremental improvement or a major advance over existing approaches.
3.  Lines 63-64: the choice of Campbell Scientific loggers and Iridium Certus 100 telemetry is stated but not justified, and the limitations of alternative systems are not discussed.

**Study site and data specification**
1.  Why were two different data loggers (CR1000 and CR1000X) used, and did their performance differ in any meaningful way?
2.  Is 12 m depth driven by scientific needs, instrumentation constraints, or power/telemetry considerations?
3.  The abstract and introduction emphasise power and data management strategies, but this section does not yet explain how the power system was sized or optimised for continuous operation.
4.  The description of the RockREMOTE Mini integration is minimal; it is unclear whether any custom configuration or scripting was required, whether any default manufacturer settings were sufficient, or any communication issues arose during integration.

**Data logging and telemetry system**

1. Line 116: CR1000/CR1000X loggers- the implications are not discussed, did their performance differ, and was this a deliberate comparison or driven by equipment availability?
2. Line 179: the rationale for choosing IMT over IP-based communication is not discussed beyond general bandwidth efficiency.
3. Line 204: Power management via the modem sleep pin is central to the design, yet no quantitative estimate of power savings is provided.
4. Line 208-209: Retry logic is well described, but the manuscript does not report how often retries were required or how effective they were in practice.
5. Line 215: Buffering data for up to 4 days is a strong feature, but it is unclear how often buffering was used, whether truncation occurred, or whether the 4-day limit is a design choice or a hard constraint.

**Power management**

1. Line 244: Power management is framed as a 'critical design challenge', yet no quantitative energy budget is presented (e.g. daily generation vs. consumption), making it difficult to assess how close the system operated to power limits.
2. The use of lead-acid batteries is stated but not discussed critically. At this altitude and temperature range:
   a. Was reduced battery efficiency or cold-temperature degradation observed?
   b. Were battery temperatures monitored, and if so, how did they relate to voltage stability?

**Interface: Cloudloop IoT platform**
1. The section clearly explains how Cloudloop displays and forwards data, but remains largely descriptive. It would be helpful to have some quantitative performance metrics, such as message delivery success rates, transmission-to-receipt delays, or frequency of failed or delayed messages.
2. It is unclear whether Cloudloop provides automated handling of missing or corrupted messages, or whether continuous manual oversight is required.

**Automated ETL pipeline for telemetry data processing**

1. Excel file size limits are mentioned but not quantified. It is unclear how well this approach would work for higher data volumes, longer deployments, or more sensors.
2. The section does not explain how errors are handled, such as missing or corrupted emails, failed decoding, or partial data transfers.

**Discussion**
1. Section 1
   a. It highlights robustness under cloud cover and monsoon conditions, but no quantitative analysis of environmental factors affecting transmission quality (e.g., snow accumulation, ice movement, temperature fluctuations) is shown.

b. The comparison with alternative loggers and low-cost platforms is informative, but a more critical evaluation of why the CR1000/CR1000X was preferred (e.g., power efficiency, programming flexibility, telemetry integration) would strengthen the discussion.

c. Similarly, the discussion of alternative satellite platforms is thorough, but mostly descriptive; including comparative performance data or justification for selecting RockREMOTE Mini/Iridium beyond coverage and low power would enhance the argument.

d. The discussion notes that data remain stored in cloud and local systems, which is a strength, but there is no critical assessment of long-term data integrity, backup strategies, or potential cloud service limitations.

2. Section 2

a. Claims that the system could support a broad range of alternative payloads are reasonable, but the manuscript does not critically evaluate limitations in power, bandwidth, or data handling that may arise with higher-volume sensors.

b. The potential for contribution to early warning systems is intriguing, but the discussion is speculative; it would benefit from concrete examples or estimated performance requirements needed to make this feasible. Are there any anticipated limitations in scaling the system for real-time hazard monitoring networks?

c. It emphasizes modularity and scalability, but does not discuss network management, cumulative power demand, or telemetry scheduling for multi-node future deployments.

**Summary remarks**

1. The summary effectively restates the main achievements of the system, emphasising reliability, power management, communication efficiency, and reproducibility.

2. Brief statements would be useful regarding future work related to:

a. How transferable are the workflows to other telemetry platforms or non-Windows environments?

b. Are there any plans to benchmark this system against alternative telemetry/logging platforms in terms of cost, reliability, or power efficiency?

---

## Author Comment (AC1)

We thank the reviewers for investing their time and providing constructive comments on our manuscript. Overall, we revised the manuscript according to their suggestions. Below, we explain the changes we have made and present our reasoning for the suggestions we didn't follow. We hope these revisions are satisfactory and the revised manuscript is deemed acceptable for publication. Our responses are tabbed and in blue and follow individual comments. The line numbers we refer to are those of the marked-up manuscript.

**RC1: 'Comment on egusphere-2025-4346', 28 Dec 2025**

**Summary:** The paper presents an integrated data logging and satellite telemetry in extreme high-altitude conditions, combining Campbell data loggers with the RockREMOTE modem. While the introduction effectively motivates the need for environmental data, and the manuscript provides detailed descriptions of system components, programming, and automated data workflows, the novelty and technical justification for design/instrument choices are often unclear. Key limitations include the lack of quantitative evaluation of telemetry performance, power consumption, data transmission reliability, and error handling. Also, minimal critical assessment of alternative loggers or satellite platforms. Overall, the manuscript demonstrates a functional and thoughtfully engineered deployment with promising automation and power management strategies for remote environmental monitoring.

We are thankful to the reviewer for the thorough read and constructive comments. We appreciate the recognition of the functional deployment and automation strategies. The manuscript has been revised after taking all the suggestions into consideration, with particular focus on adding quantitative performance metrics, critical evaluation of alternatives, and clearer justification of design choices.

**Abstract**

1. The title is generic and does not indicate what aspect of the system is novel.

   Thank you for this point. We have revised the title to better reflect the specific contribution of our work.

   Revised title: *Design and implementation of a robust data logging and satellite telemetry system for remote cryospheric research*

2. The motivation in the abstract does not explain what exact limitations in existing data logging or telemetry systems necessitated this work.

   We have revised the abstract to explicitly state the limitations being addressed.

   Details are added in the revised manuscript as follows:

   L17-22: *Scientific research in remote environments has traditionally relied on manual data retrieval from data loggers, requiring multiple field visits that are costly, logistically challenging, and sometimes hazardous. While satellite telemetry solutions exist, their integration with widely used research-grade data loggers in extreme environments remains poorly documented, limiting reproducibility and adoption.*

3. It implies novelty using recent satellite IoT connectivity but does not explain why this integration represents an advance over previously available telemetry solutions. If this is the first time IoT has been used in this research domain, it would be helpful to highlight it.

*Thank you for the suggestion. We have added clarification in the abstract to highlight the novelty of this integration.*

*Details are added in the revised manuscript as follows:*

*L30-32: This work represents the first documented integration of research-grade Campbell Scientific loggers with Iridium Certus 100 satellite IoT technology, providing validated protocols, performance metrics, and automated workflows for extreme environments.*

**Introduction**

1. Lines 51-60: it does not clearly state a specific technical problem it addresses (e.g. data transmission reliability, power limits, data loss, or system lifetime), instead describing a general need for remote monitoring.

*Thank you for the suggestion. We have revised the Introduction to more explicitly link telemetry developments to specific scientific questions in mountain climate and glacier research.*

*Details are added in the revised manuscript as follows:*

*L67-69: Near-real-time telemetry directly supports investigation of glacier-climate interactions in high-mountain and polar environments by enabling continuous retrieval of mission-critical environmental data.*

2. The literature focuses on field logistics and maintenance, with little discussion of existing satellite telemetry systems, making the novelty of the work unclear. It is not evident whether the system is an incremental improvement or a major advance over existing approaches.

*We have added a new paragraph reviewing existing telemetry approaches and positioning our contribution.*

*Details are added in the revised manuscript as follows:*

*L70-79: Existing satellite telemetry deployments in glaciology have primarily fallen into three categories (1) custom-integrated commercial systems combining sensors and telemetry in proprietary packages, which offer reliability but limit sensor flexibility and require notable capital investment; (2) purpose-built open-source platforms designed around specific satellite protocols (Garbo & Mueller, 2024), which provide transparency but require electronics expertise and may not meet the precision requirements of all applications; and (3) research-grade logger deployments with aftermarket satellite additions, which remain largely undocumented thus far. Our deployment addresses the third category by providing full documentation of a validated integration between Campbell Scientific CR1000/CR1000X loggers and the RockREMOTE Mini using Iridium Certus 100 connectivity.*

3. Lines 63-64: the choice of Campbell Scientific loggers and Iridium Certus 100 telemetry is stated but not justified, and the limitations of alternative systems are not discussed.

We have added explicit justification. Alternative platforms are critically evaluated in Section 7.1.

Details are added in the revised manuscript as follows:

L466-477: *The CR1000(X) platform was selected for four primary reasons (1) proven reliability in extreme environments, with documented operation from −40 °C to +70 °C; (2) compatibility with a wide range of research-grade meteorological and thermistor sensors required for cryospheric studies; (3) programming flexibility through the CRBasic environment, enabling custom power management, conditional sampling, retry logic, and direct AT-command control of satellite modems; and (4) extensive community adoption and institutional support, reducing development and operational risk. Compared with low-cost platforms such as Raspberry Pi or Arduino (Nazir et al., 2017; Chan et al., 2021), the CR1000(X) provides a favourable balance between power efficiency and advanced functionality, including native RS-232 interfaces, precise timing, programmable digital I/O, and support for differential analogue measurements without external signal conditioning. However, alternative data acquisition systems may be appropriate under different application-specific constraints. Other platforms,.....*

L485-491: *The RockREMOTE Mini was chosen based on (1) ultra-low power consumption (300 mW operating, enabling solar-battery systems); (2) Iridium Certus 100 compatibility, providing truly global coverage including polar regions; (3) an omnidirectional antenna, eliminating pointing requirements in dynamic remote environments; (4) serial interface compatibility with CR1000(X), enabling direct integration without protocol conversion; and (5) message-based IMT protocol support, allowing efficient low-bandwidth transmission (up to 100 KB per message) optimised for environmental monitoring.*

**Study site and data specification**

1. Why were two different data loggers (CR1000 and CR1000X) used, and did their performance differ in any meaningful way?

Two different logger models were deployed, the CR1000 for the AWS and CR1000X for the two thermistor strings. This choice was driven by equipment availability rather than a deliberate performance comparison, as both models share identical core functionality for our application. The CR1000X offers enhanced memory (72 MB vs. 4 MB) and faster processor (100 MHz vs. 32 MHz), but these advantages were not exploited in our deployment. Both models demonstrated equivalent performance throughout the operation period, with no measurable differences in data quality, power consumption, or telemetry success rates. For simplicity, we refer to both as CR1000/CR1000X throughout this manuscript, noting that either model alone would be sufficient for similar deployments.

Details are added in the revised manuscript as follows:

L145-146: *While two different logger models were used, both share identical core functionality for our application and demonstrated equivalent performance throughout the deployment.*

2. Is 12 m depth driven by scientific needs, instrumentation constraints, or power/telemetry considerations?

The 12 m borehole depth was determined by scientific objectives and logistic constraints rather than power or telemetry limitations. This depth captures seasonal temperature variability within the active firn layer while extending into the zone where annual temperature fluctuations are strongly damped, providing critical information on energy penetration and subsurface heat storage (Miles et al., 2018). The thermistor string configuration was optimised to resolve vertical thermal gradients while maintaining efficient data handling. In our deployment, the thermistor dataset transmitted daily is a modest ~7 KB per transmission, well below the system's telemetry capacity. Power and telemetry considerations therefore did not constrain sensor deployment. Had scientific requirements necessitated deeper installations or higher spatial resolution, the system could accommodate substantially larger data volumes (up to 100 KB per message) through adjustable transmission intervals or data compression strategies.

3. The abstract and introduction emphasise power and data management strategies, but this section does not yet explain how the power system was sized or optimised for continuous operation.

The focus of this section is deliberately limited to the study site and data specifications, and therefore it does not include details of power system sizing or optimisation. These aspects are addressed explicitly in Section 4 (Power management), where the power system design, sizing rationale, and optimisation for continuous operation are described in detail.

4. The description of the RockREMOTE Mini integration is minimal; it is unclear whether any custom configuration or scripting was required, whether any default manufacturer settings were sufficient, or any communication issues arose during integration.

Thank you for your suggestion. No custom firmware modifications were required for either component. All configuration was achieved through CRBasic programming and standard AT commands.

Details are added in the revised manuscript as follows:

L169-172: *The physical integration required three connections: (1) GPIO cable for power (12V from logger) and serial communication, (2) LMR 400 cable to the externally mounted antenna, and (3) Ethernet cable for optional Power over Ethernet (PoE) (not used in our deployment).*

**Data logging and telemetry system**

1. Line 116: CR1000/CR1000X loggers- the implications are not discussed, did their performance differ, and was this a deliberate comparison or driven by equipment availability?

Addressed above in 'Study Site' response 1. The model selection was driven by availability rather than performance requirements, with both loggers demonstrating equivalent operational characteristics.

2. Line 179: the rationale for choosing IMT over IP-based communication is not discussed beyond general bandwidth efficiency.

Thank you for this constructive comment. Beyond bandwidth efficiency, IMT's store-and-forward architecture enables reliable data delivery under intermittent satellite visibility. IMT buffers complete messages on the transceiver and transmits them in small segments, if visibility is lost, transmission resumes automatically when the link is restored. IMT can also operate at lower signal levels (–117 to –119 dBm) than IP (which typically fails below ~–115 dBm), making it better suited to obstructed sky conditions. The main limitations of IMT are its unidirectional nature and fixed message size limit (≤100 KB). By contrast, IP connectivity enables two-way communication and higher data volumes but at higher power and cost.

Details added to the revised manuscript are as follows,

L185-195: *The IMT was selected over IP-based connectivity because its store-and-forward architecture is robust to the intermittent satellite visibility encountered at the field site. Complete messages are buffered within the transceiver and transmitted in small segments, resuming automatically if the link is interrupted. IMT remains operational at received signal levels of approximately -117 to -119 dBm, whereas typical IP-based connectivity operates over a stronger signal range of approximately -110 to -114 dBm and generally fails below -115 dBm, providing a 2-4 dB tolerance under weak-signal or obstructed conditions. These characteristics enable reliable one-way telemetry under low-power conditions. The limitations of IMT include a fixed message size limit (≤100 KB), meaning that applications requiring two-way communication, remote configuration, or larger data volumes may instead favour IP connectivity despite higher energy and cost requirements.*

3. Line 204: Power management via the modem sleep pin is central to the design, yet no quantitative estimate of power savings is provided.

Thank you for the suggestion. We have now added a quantitative analysis of the power savings achieved through modem sleep-pin management.

L255-259: *With sleep mode enabled, baseline power consumption is reduced to <36 mW at 12 V (~75 mW at 24 V), with the Iridium 9770 transceiver drawing power primarily during scheduled transmission periods. Without sleep-pin management, continuous modem operation would dominate total power use, requiring larger batteries or substantially larger solar arrays (~50–75 W compared with the 30 W array deployed).*

4. Line 208-209: Retry logic is well described, but the manuscript does not report how often retries were required or how effective they were in practice.

This information is already reported in Section 7.1 (System performance and reliability), where we quantify transmission success rates, the frequency of retry attempts, and their effectiveness based on the full deployment record (see also Fig. 9).

5. Line 215: Buffering data for up to 4 days is a strong feature, but it is unclear how often buffering was used, whether truncation occurred, or whether the 4-day limit is a design choice or a hard constraint.

   The four-day buffering period is a design choice determined by datalogger internal memory allocation, not by the satellite communication system. In our CR1000/CR1000X program, string variables were limited to 25,000 characters. With 30-minute sampling (48 records per day) and ~6,000 characters per day (~7 kB), this allows buffering of approximately four days (~24,000 characters). No truncation occurred during the deployment. This limit is user-configurable and can be increased by allocating additional logger memory or reducing per-record string length. By contrast, the RockREMOTE Mini using Iridium Certus 100 IMT supports message sizes up to 100 kB (~100,000 characters), which was not a limiting factor in this deployment.

**Power management**

1. Line 244: Power management is framed as a 'critical design challenge', yet no quantitative energy budget is presented (e.g. daily generation vs. consumption), making it difficult to assess how close the system operated to power limits.

   Thank you for this comment. We have now included a quantitative estimate of power consumption. These values were measured under worst-case signal conditions and maximum cable lengths; under more favourable conditions, power consumption is expected to be lower. Incorporating these measurements allows assessment of the system's energy demands relative to battery and solar capacity and demonstrates that power management strategies, particularly sleep-pin control, were essential to maintain continuous operation.

   L175-180: *The RockREMOTE's low operating power consumption of <36 mW in sleep mode and ~300 mW in idle mode supports efficient field operation. During Iridium transmissions, the unit draws an average of ~7,250 mW, with absolute peaks up to ~20,000 mW during satellite uplink bursts. The availability of a sleep pin for dynamic power management makes the system ideally suited to solar-powered or battery-constrained deployments in remote environments.*

2. The use of lead-acid batteries is stated but not discussed critically. At this altitude and temperature range:
   a. Was reduced battery efficiency or cold-temperature degradation observed?

   We observed temperature-induced capacity loss in the lead-acid batteries during the observation period. During daytime, when solar insolation was present, the batteries were in charging mode and exhibited higher voltages. At night, as temperatures decreased in the absence of sunlight, a voltage drop was observed. Over the observation period, the average temperature was -6.3 °C, with substantial variability ranging from a minimum of -16.4 °C to a maximum of 4.1 °C. However, we cannot definitively quantify the effects of temperature without controlled testing, so for that reason we have stopped short of revising the manuscript in this regard. For deployments in environments with

temperatures below -20 °C, lithium batteries with low-temperature specifications would be strongly preferable.

b. Were battery temperatures monitored, and if so, how did they relate to voltage stability?

Battery temperatures were not directly monitored. Consequently, we cannot directly relate battery temperature to voltage stability.

**Interface: Cloudloop IoT platform**

1. The section clearly explains how Cloudloop displays and forwards data, but remains largely descriptive. It would be helpful to have some quantitative performance metrics, such as message delivery success rates, transmission-to-receipt delays, or frequency of failed or delayed messages.

Cloudloop is the platform provided by Ground Control through which we received the transmitted data. As such, we do not have access to internal performance metrics such as message delivery success rates, transmission-to-receipt delays, or frequency of failed or delayed messages. Our observations are limited to the data successfully received via Cloudloop, which, during the deployment period, showed no evidence of missing or corrupted messages.

2. It is unclear whether Cloudloop provides automated handling of missing or corrupted messages, or whether continuous manual oversight is required.

Cloudloop provides automated logging of message receipt but does not implement active error correction or automated retries for missing data. The platform's role is passive, it receives messages from RockREMOTE Mini and forwards them via configured delivery methods (email in our case).

There are three implications that arise from this:

1. Missing or corrupted messages: Error checking occurs between the RockREMOTE Mini and the Iridium satellite gateway via Cyclic Redundancy Checks (CRC), meaning corruption is unlikely to arise within the satellite link itself. In principle, corruption could only occur upstream between the data logger and the modem, although mechanisms such as IMTWB can optionally apply CRC at this stage. However, if a corrupted or malformed payload were to reach Cloudloop, the platform would not detect or correct it and would forward the packet unchanged. Similarly, if a transmission fails before reaching Cloudloop, the platform has no awareness of the missing message and cannot trigger recovery, responsibility for retransmission resides entirely with the field system.

2. Corrupted messages: While we observed no corruption events, the platform would forward corrupted payloads unchanged. Data validation must occur downstream (in our case, through Excel formulas checking for expected timestamp formats and value ranges).

3. Manual oversight: Our manual checks consisted solely of verifying receipt of emails from Cloudloop, there was no need to access or monitor the Cloudloop platform itself.

While not strictly necessary for system operation (automated workflows continue regardless), this simple oversight proved valuable for detecting sensor issues or transmission failures.

For fully autonomous operation without manual oversight, users should implement automated data validation and anomaly detection in their downstream processing (e.g., Python scripts checking for missing timestamps, out-of-range values, or sensor drift patterns).

**Automated ETL pipeline for telemetry data processing**

1. Excel file size limits are mentioned but not quantified. It is unclear how well this approach would work for higher data volumes, longer deployments, or more sensors.

   We thank the reviewer for this comment. We have now added quantitative detail on Excel's relevant file size and structural limits to clarify scalability

   L383-389: *Following this ETL sequence, we implemented Power Automate within MS Excel to automate reading the updated text file, converting the HEX encoded content into ASCII format, and loading the decoded data into a structured dataset for analysis. Excel has significant structural and performance limits, most notably a maximum of 1,048,576 rows and 16,384 columns per worksheet, with individual cells capable of holding up to 32,767 characters. Practical limits also apply to total file size, which becomes memory- and system-dependent as files grow.*

2. The section does not explain how errors are handled, such as missing or corrupted emails, failed decoding, or partial data transfers.

   We have expanded the section to describe how the automated workflow handles data errors and incomplete transmissions.

   Details added to the revised manuscript are as follows:

   L373-383: *The Power Automate workflow implements multiple error-handling mechanisms. (1) Missing emails: The workflow is event-driven and triggers only upon email arrival, it does not poll for expected transmissions. If a scheduled transmission does not arrive, the system continues operating normally, with the gap becoming apparent during routine quality control. (2) Malformed emails: The workflow checks for the presence of HEX payload markers. Emails lacking valid markers are skipped and treated as data gaps rather than causing system failure. (3) HEX decoding failures: In Excel, the decoding formulas verify that payloads contain valid hexadecimal pairs (0–9, A–F). Payloads failing this check are discarded, preventing propagation of corrupted values into the data pipeline. In all error scenarios, original Cloudloop messages remain accessible through the web interface, providing a recovery pathway for manual retrieval if required.*

**Discussion**

1. Section 1

a. It highlights robustness under cloud cover and monsoon conditions, but no quantitative analysis of environmental factors affecting transmission quality (e.g., snow accumulation, ice movement, temperature fluctuations) is shown.

Environmental conditions can influence transmission reliability primarily through mechanical obstruction or displacement rather than through atmospheric effects. Snow accumulation is the dominant constraint for surface-deployed systems. If an omnidirectional antenna becomes partially or fully buried, the L-band signal can be attenuated or entirely blocked. Because omnidirectional antennas only require a vertical orientation, small tilts typically have negligible impact on link quality; however, complete mast failure or a transition to a horizontal orientation would prevent successful transmissions until the antenna is restored.

Temperature is generally not a limiting factor, as RockREMOTE modem is specified for operation across a wide temperature range (-40 °C to +70 °C). Similarly, common atmospheric variables such as humidity, wind speed, or air pressure have no documented effect on transmission success for this class of systems. Surface motion (e.g., ice flow) may introduce long-term mechanical risks by altering antenna orientation or stressing cabling, but short-term motion at typical glacier velocities is unlikely to affect satellite link performance.

Taken together, the main environmental hazards for field deployments are physical burial or displacement rather than meteorological variability, and system design should prioritise antenna elevation, structural robustness, and inspection accessibility.

b. The comparison with alternative loggers and low-cost platforms is informative, but a more critical evaluation of why the CR1000/CR1000X was preferred (e.g., power efficiency, programming flexibility, telemetry integration) would strengthen the discussion.

We thank the reviewer for this constructive suggestion. We have expanded the Discussion to provide a critical evaluation of why the CR1000/CR1000X was selected over alternative data logging platforms, with particular focus on power efficiency, programming flexibility, and telemetry integration. Addressed as above in 'Introduction' response 2 (L70-79).

c. Similarly, the discussion of alternative satellite platforms is thorough, but mostly descriptive; including comparative performance data or justification for selecting RockREMOTE Mini/Iridium beyond coverage and low power would enhance the argument.

We have included comparative performance data and detailed justification for selecting the RockREMOTE Mini/Iridium Certus 100 platform over alternative satellite communication platforms.

Details added to the revised manuscript are as follows:

L495-503: *Further, the current deployment relied on the RockREMOTE Mini modem operating over the Iridium satellite network due to its global coverage and low power*

*demand, but also for its favourable balance between power consumption, supported message payload size, operational robustness, and tolerance to restricted sky visibility. In addition to global coverage, the RockREMOTE Mini with Iridium Certus 100 supports store-and-forward operation, short transmission windows, and single-message payloads of up to 100 KB, enabling fully automated daily data delivery with minimal latency under constrained power budgets and intermittent satellite visibility conditions that are difficult to achieve simultaneously in geostationary or broadband systems.*

L515-526: *Relative to other telemetry systems, the Iridium Certus 100 link supports modest data volumes with short transmission durations and high delivery reliability, enabling consistent daily messaging without continuous sky visibility. The compact omnidirectional antenna eliminates pointing requirements and reduces mechanical failure modes, while the wide operational temperature range enables deployment in cold environments without additional environmental conditioning. Together with aggressive duty-cycle power management and a modular system architecture, these characteristics represent a substantial improvement over many previously reported telemetry deployments, for which power consumption, latency, or transmission reliability are often described only qualitatively. As a result, low-power Iridium-based systems provide a robust and operationally efficient solution for remote cryospheric and high-mountain deployments where power availability, sky visibility, and site accessibility are the dominant constraints.*

d. The discussion notes that data remain stored in cloud and local systems, which is a strength, but there is no critical assessment of long-term data integrity, backup strategies, or potential cloud service limitations.

Thank you for this comment. Cloudloop is built on AWS and uses MQTT for message handling, with modern data integrity mechanisms and up-to-date security certificates. Data are stored indefinitely by default unless a customer explicitly modifies retention settings. Messages from the Iridium satellite network are received at the Tempe, Arizona Ground Station, forwarded via Ground Control's simple queue service for IMT messages, and for IP-based transmissions, sent via MPLS to Ground Control's NYC point-of-presence (POP), which features two physical servers and four routers (two each at NYC and Tempe) with dual independent MPLS circuits. Cloudloop maintains high reliability, with 100% uptime reported on https://status.groundcontrol.com/, and background message forwarding tasks continue to operate even if the web portal is unavailable. These measures collectively ensure robust long-term data integrity, redundancy, and operational reliability.

Details added to the revised manuscript are as follows:

L324-334: *It provides robust long-term data storage and integrity, and is built on AWS and uses Message Queuing Telemetry Transport (MQTT), with modern data integrity mechanisms and up-to-date security certificates. By default, data are retained indefinitely unless a customer modifies retention settings. Iridium messages are received at the Tempe, Arizona Ground Station and routed via Ground Control's SQS queue (IMT) or*

*MPLS (IP) to Ground Control's NYC POP, which features redundant servers and routers across both sites with dual independent MPLS circuits. Cloudloop maintains 100% uptime (https://status.groundcontrol.com/, last access: 6 February 2026), and background message forwarding continues even if the web portal is unavailable. These features ensure reliable, redundant storage and continuous data delivery, minimising the risk of data loss or service interruption.*

2. Section 2

   a. Claims that the system could support a broad range of alternative payloads are reasonable, but the manuscript does not critically evaluate limitations in power, bandwidth, or data handling that may arise with higher-volume sensors.

   Thank you for this comment. We have added a discussion of the limitations associated with higher-volume sensors on the RockREMOTE Mini.

   L599-610: *The suitability of the RockREMOTE Mini for alternative payloads depends on power availability, data volume, and transmission frequency. While the modem supports a variety of sensor types, high-volume or high-frequency data streams are constrained by practical limits in power consumption, bandwidth, and onboard buffering. The Iridium 9770 transceiver supports uplink transmissions of up to 22 Kbps and downlink transmissions of up to 88 Kbps via IP, or up to 100 KB in each direction using the IMT protocol. Messages are queued in an internal buffer capable of storing approximately 6,000 messages (~3 MB total), with a smaller immediate buffer accommodating roughly 100 KB outgoing and 35 KB incoming data simultaneously, of which ~15 KB is reserved for system overhead. These constraints make the system well suited to low- to moderate-data-rate sensors, while applications generating large data volumes or requiring frequent transmissions would require additional data reduction, compression, or alternative telemetry solutions.*

   b. The potential for contribution to early warning systems is intriguing, but the discussion is speculative; it would benefit from concrete examples or estimated performance requirements needed to make this feasible. Are there any anticipated limitations in scaling the system for real-time hazard monitoring networks?

   We have expanded the discussion to provide concrete examples, performance requirements, and limitations for scaling the system to real-time hazard monitoring networks.

   Details added to the revised manuscript are as follows:

   L613-624: *With targeted modifications, such as higher-frequency or threshold-triggered transmissions, the system could support early warning applications for a range of environmental and geotechnical hazards, including glacial lake outburst floods, slope instability, permafrost degradation, extreme melt events, and hydrometeorological extremes. The modularity of the system makes it adaptable for regional or national monitoring frameworks, enabling scalable networks that can provide timely data to both researchers and decision-makers across cryospheric, marine, terrestrial, and*

*atmospheric monitoring domains, where performance requirements are typically defined by reliable near-real-time updates (on the order of hours to one day), high transmission reliability, and sustained operation under constrained power budgets rather than continuous high-bandwidth data streams. Scaling such networks is constrained by power availability, transmission frequency, and operational costs, and applications requiring sub-hourly updates or continuous data transmission.*

c. It emphasizes modularity and scalability, but does not discuss network management, cumulative power demand, or telemetry scheduling for multi-node future deployments.

We thank the reviewer for this comment. While the manuscript highlights the system's modularity and potential for multi-node deployments, detailed discussion of network management, cumulative power demand, or telemetry scheduling for large-scale networks is beyond the scope of the current study and is suggested as a topic for future work.

**Summary remarks**

1. The summary effectively restates the main achievements of the system, emphasising reliability, power management, communication efficiency, and reproducibility.

Thank you for the positive feedback. We appreciate the recognition.

2. Brief statements would be useful regarding future work related to:
a. How transferable are the workflows to other telemetry platforms or non-Windows environments?

Thank you for the suggestion. Details added to the revised manuscript are as follows:

L648-657: *The workflows presented are primarily implemented on Windows-based Power Automate and Excel environments. While the underlying concepts (automated email retrieval, data parsing, and quality control) are transferable, adaptation to alternative telemetry platforms or non-Windows operating systems would require custom scripting or equivalent automation tools. Comparable workflows can be implemented using programming-based solutions or other automation platforms, depending on user preference and expertise. While Excel-based processing was adopted here to maximise accessibility and enable rapid deployment, we recommend that operational deployments adopt programming-based workflows (e.g. Python or R), which provide more flexible and powerful alternatives, including improved error handling, version control, scalability, and reproducibility.*

b. Are there any plans to benchmark this system against alternative telemetry/logging platforms in terms of cost, reliability, or power efficiency?

Systematic benchmarking against alternative telemetry or logging platforms is not included in the current study and is proposed as future work to inform broader application and platform selection.

***References***

Garbo, A., & Mueller, D.: Cryologger Ice Tracking Beacon: A Low-Cost, Open-Source Platform for Tracking Icebergs and Ice Islands. Sensors 2024, Vol. 24, Page 1044, 24(4), 1044, https://doi.org/10.3390/S24041044, 2024.

Miles, K. E., Hubbard, B., Quincey, D. J., Miles, E. S., Sherpa, T. C., Rowan, A. V., & Doyle, 540 S. H.: Polythermal structure of a Himalayan debris-covered glacier revealed by borehole thermometry. Scientific Reports, 8(1), https://doi.org/10.1038/s41598-018-34327-5, 2018.